# In Situ Vitrification of Lung Cancer Organoids on a Microwell Array

**DOI:** 10.3390/mi12060624

**Published:** 2021-05-28

**Authors:** Qiang Liu, Tian Zhao, Xianning Wang, Zhongyao Chen, Yawei Hu, Xiaofang Chen

**Affiliations:** 1Department of Thoracic Surgery, Beijing Haidian Hospital, Beijing 100080, China; drlaoliu@sina.com; 2Beijing OrganoBio Corporation, Beijing 102206, China; zhaotian58685@gmail.com (T.Z.); wangxianning191211@gmail.com (X.W.); 3Department of Biomedical Engineering, School of Medicine, Tsinghua University, Beijing 100084, China; Czy18@mails.tsinghua.edu.cn (Z.C.); huyw@tsinghua.edu.cn (Y.H.); 4Key Laboratory for Biomechanics and Mechanobiology of Ministry of Education, School of Biological Science and Medical Engineering, Beihang University, Beijing 100083, China

**Keywords:** patient-derived organoid, cryopreservation, in situ vitrification, high throughput screening

## Abstract

Three-dimensional cultured patient-derived cancer organoids (PDOs) represent a powerful tool for anti-cancer drug development due to their similarity to the in vivo tumor tissues. However, the culture and manipulation of PDOs is more difficult than 2D cultured cell lines due to the presence of the culture matrix and the 3D feature of the organoids. In our other study, we established a method for lung cancer organoid (LCO)-based drug sensitivity tests on the superhydrophobic microwell array chip (SMAR-chip). Here, we describe a novel in situ cryopreservation technology on the SMAR-chip to preserve the viability of the organoids for future drug sensitivity tests. We compared two cryopreservation approaches (slow freezing and vitrification) and demonstrated that vitrification performed better at preserving the viability of LCOs. Next, we developed a simple procedure for in situ cryopreservation and thawing of the LCOs on the SMAR-chip. We proved that the on-chip cryopreserved organoids can be recovered successfully and, more importantly, showing similar responses to anti-cancer drugs as the unfrozen controls. This in situ vitrification technology eliminated the harvesting and centrifugation steps in conventional cryopreservation, making the whole freeze–thaw process easier to perform and the preserved LCOs ready to be used for the subsequent drug sensitivity test.

## 1. Introduction

Tumor cell lines have been used worldwide as primary tools for anti-cancer drug development due to their relevance to cancers (i.e., mutations in oncogenes), unlimited proliferation capacities, and well-developed high-throughput culture and analysis systems from multi-well plates to liquid handling robots. However, cell lines cannot resemble the three-dimensional (3D) structure and the heterogeneity of real tumor tissues, leading to differences in drug responses between cell lines and in vivo models. In recent years, patient-derived organoids (PDOs) have attracted much attention due to their similarity to in vivo tumor tissues and are recognized as a promising in vitro model to fill the gap between cell lines and in vivo models. PDOs are self-organized three-dimensional cultures of patient tumor cells, retaining the 3D structure and genetic mutations of the parental tumor tissues [1]. PDOs can be established from many different types of tumor tissues, including colorectal cancers [2], breast cancers [3], lung cancers [4], ovary cancers [5], etc. Previous reports in colorectal cancer organoids demonstrated that PDOs captured patient’s responses to anti-tumor therapies [6,7]. PDO-based drug candidate validation has been explored and promising results were reported [8,9,10].

Although the potentials of PDOs have been recognized, suitable culture and analysis systems have to be developed to facilitate the application of PDOs in cancer research and anti-tumor drug development. Firstly, culturing and manipulating PDOs are more complex and expensive than that of the cell lines due to the requirement of the 3D culture matrix. Secondly, PDOs show tremendous diversities in genetic mutation, morphology, and proliferation potency due to the heterogeneity of the parental tumors. In addition, the proliferation capacity of tumor organoids is limited compared to cell lines. For instance, it has been reported that lung cancer organoids (LCOs) were overtaken by normal cells in long-term culture at a high frequency [11]. Our study also found that lung cancer organoids stopped growing at high passages. Although the proliferation of LCOs can be improved by optimizing the culturing conditions, a robust cryopreservation technology compatible with organoid culture and analysis will facilitate the usage of PDOs in anti-cancer drug development.

To date, cryopreservation strategies are mainly divided into the slow freezing and vitrification methods [12]. The most common and traditional slow freezing techniques feature a low concentration of cryoprotectants (CPA) and a slow cooling rate, which usually needs to be optimized for different cell types. Thus far, most of the reported biobanks of PDOs have been cryopreserved using the slow freezing methods [13]. On the contrary, the vitrification method employs high concentrations of CPA together with an extremely fast rate of cooling. Owing to the advantage of ice-free solution in the process of freezing, the vitrification method is regarded as the most promising way to achieve organ cryopreservation in the future [14]. Vitrification of cancer organoids has also been investigated in recent years [15,16,17,18] and has shown promising results.

In our previous studies, we developed the superhydrophobic microwell array chip (SMAR-chip) [19,20,21] and demonstrated the feasibility of PDO culture and analysis on the SMAR-chip [22]. Owning to the nanoliter-scale culture volume on the SMAR-chip, the requirement for the number of PDOs is largely reduced comparing to conventional multi-well plates. In order to facilitate the high-throughput PDO-based drug testing on the SMAR-chip, here we developed an in situ vitrification method to freeze the LCOs on the SMAR-chip using simple procedures. We proved LCOs frozen on the chip had similar viability and growth rate as those frozen in conventional cryovials. More importantly, the freeze–thaw process did not affect the responses of the organoids to anti-cancer drugs. The in situ cryopreservation together with the subsequent high-throughput drug sensitivity analysis provide a promising platform for the future application of PDOs in anti-cancer drug development.

## 2. Materials and Methods

### 2.1. The Fabrication of the SMAR-Chip

The SMAR-chip was fabricated by casting a layer of superhydrophobic paint on the top of the polycarbonate microwell array-chip [22]. The chip was manufactured by standard injection molding by Mudu Qingyuan (Jiangsu, China). The superhydrophobic paint was prepared following Lu’s protocol [23]. Briefly, 1 g of 1H, 1H, 2H, 2H-perfluorooctyltriethoxysilane (Sigma-Aldrich, St. Louis, MO, USA) was added into 99 g of absolute ethanol and mechanically stirred for 2 h. Then, 6 g of titanium oxide (TiO_2_) nanoparticles (~60 to 200 nm) (Sigma-Aldrich, St. Louis, MO, USA) and 6 g of P25 TiO_2_ (~21 nm) (Degussa, Essen, Germany) were added into the solution to make a paint-like suspension. The paint was then pipetted onto the top surface of the microwell array chip into the spaces between the microwells and air-dried completely. The SMAR-chip was autoclaved before use.

### 2.2. Culture, Passaging and Harvesting of Lung Cancer Organoids

To culture LCOs in a multi-well plate, LCOs in suspension were first centrifuged for 5 min at 500× *g* at 4 °C and resuspended in pre-cooled (4 °C) Matrigel (BD Biosciences, San Jose, CA, USA). Then, 50 μL drops of the organoid suspension were inoculated in 24-well plates and allowed to solidify at 37 °C for 20 min. The seeding density was adjusted to approximately 500 organoids per well. Subsequently, 600 μL of LCO culture medium (LCOM) was added into the wells and the plate was transferred to a cell culture incubator at 37 °C with 5% CO_2_. A detailed recipe of LCOM can be seen in Appendix A. The culture medium was replenished every 3 days. To harvest the LCOs, the culture medium was removed and 10× volumes of cold Organoid Harvesting Solution (R&D Systems, Minneapolis, MN, USA) were added into each well. The plate was shaken on an orbital shaker at 0 °C for 2 h to dissolve the Matrigel. Once the Matrigel was digested completely, the organoid suspension was sheared by pipetting, followed by washing with Advanced DMEM/F12, and centrifugation (500× *g*, 5 min, 4 °C) to collect the LCOs.

For the on-chip organoid culture, 0.4 μL of Matrigel solution containing 3–5 organoids was loaded into each microwell with an electronic pipette (Rainin E4 XLS, Mettler-Toledo, Columbus, OH, USA) working in a low-speed multi-dispense mode. Each Matrigel droplet in the microwell was overlaid with 2.4 μL of LCOM, which was replenished daily.

### 2.3. Vitrification of LCOs

For the in-vial vitrification, following 48 h of culturing, organoids were harvested, washed, and vitrified using a vitrification freeze kit (Nanjing Aibei biotech, Nanjing, China) according to manufacturer’s instructions. Briefly, the organoids were harvested and suspended in the equilibration solution for 5 min, then resuspended in vitrification reagent after centrifugation and transferred into liquid nitrogen. To thaw the LCOs, a thawing kit (Nanjing Aibei biotech, Nanjing, China) was used, following the manufacturer’s instructions. Briefly, the cryovial was removed from the liquid nitrogen and placed in a 37 °C water bath and agitated until only a pea-sized piece of ice remained. After centrifugation, organoids were mixed with thawing solution and incubated for 5 min followed by three washes with the LCOM. Then, the LCOs were mixed with Matrigel, loaded into a 24-well plate, and cultured under normal conditions.

For the in situ vitrification on the SMAR-chip, Matrigel with organoids was first loaded as described above. After 2 h of incubation in the CO_2_ incubator, 2 μL of equilibration solution (Nanjing Aibei biotech, Nanjing, China) was added on top of the Matrigel droplets and incubated for 5 min. Then, the equilibration buffer was removed with a piece of filter paper and replaced by 2 μL of vitrification solution (Nanjing Aibei biotech, Nanjing, China). After 2 min of liquid exchange, the chip was sealed and placed into liquid nitrogen. When thawing, the chip was put into a 37 °C incubator for 20 s followed by removal of the vitrification solution by gentle wiping with a piece of filter paper. Then, 2 μL of thawing solution (Nanjing Aibei biotech, Nanjing, China) was added onto the Matrigel droplets followed by three washes with the LCOM. After washing, the chip was transferred to the 37 °C incubator under normal on-chip organoid culture conditions. 

### 2.4. Slow Freezing of LCOs

Briefly, organoids were suspended in cell cryopreservation medium (CELLBANKER^TM^, ZEN OAQ, Fukushima, Japan) and transferred into cryovials. Cryovials were sealed and cooled to −80 °C in Corning CoolCell Containers (Corning, NY, USA). After 24 h, cryovials were transferred to liquid nitrogen. To thaw the LCOs, the cryovial was removed from the liquid nitrogen, placed in a 37 °C water bath, and agitated until only a pea-sized piece of ice remained. Then, 1 mL of pre-warmed LCOM was added into the cryovial and the LCOs were centrifuged at 1000 rpm for 5 min, resuspended in Matrigel, and cultured under normal conditions.

### 2.5. Evaluation of Cell Viability

Organoid viability was determined using the LIVE-DEAD cell viability kit (YEASEN biotech, Shanghai, China). Calcein AM at a concentration of 2 μM and propidium iodide at a concentration of 4 μM were added to the LCOs and incubated for 15 min, followed by imaging of the LCOs with an Olympus IX83 inverted fluorescence microscope.

### 2.6. Quantitative Real-Time Polymerase Chain Reaction

Total RNA of each sample was extracted using TRIzol Reagent (Invitrogen, Carlsbad, CA, USA) according to the manufacturer’s instructions. After that, 50 ng of RNA for each reaction was used to perform one-step RT-qPCR following the manufacturer’s instructions (Takara, Dalian, China). The reactions were performed in the CFX96 Touch Real-Time PCR Detection System (Bio-Rad, Hercules, CA, USA) with three replicates for each sample. The relative mRNA levels of the target genes were analyzed using the ΔΔCT method with the internal reference gene, GAPDH. Primers used in this reaction are listed in Appendix A.

### 2.7. Flow Cytometry Analysis

Organoids were digested into single cells with trypsin. Then, cells were fixed with a Fixation/Permeabilization Solution Kit (BD, San Jose, CA, USA) according to the manufacturer’s instructions. Briefly, 100 µL of Fixation/Permeabilization solution was mixed with resuspended cells and incubated for 20 min at 4 °C. The cells were washed twice with 1× BD Perm/Wash buffer. Then, 5 µL of Alexa Fluor 488 conjugated anti-Bcl-2 (Biolegend, San Diego, CA, USA) or anti-Bcl-XL (CST, Danvers, MA, USA) antibodies were added into the tube and incubated for 30 min at 4 °C. Stained cells were analyzed using a BD Aria SORP Flow Cytometer (San Jose, CA, USA). Results were plotted using FlowJo (LLC, Ashland, OR, USA).

### 2.8. Histology and Immunostaining

Harvested organoids were suspended in 40 μL of 10 mg/mL Fibrinogen solution (Sigma-Aldrich, St. Louis, MO, USA), and then immediately mixed with 10 units of Thrombin reagent (Solarbio, Beijing, China) for fibrin polymerization. The LCOs embedded in the fibrin hydrogel were fixed in 1 mL of 4% paraformaldehyde (Sigma-Aldrich, St. Louis, MO, USA), followed by dehydration, paraffin embedding, sectioning, and a standard H&E staining protocol. Immunohistochemistry staining was performed according to the standard immunohistochemistry staining protocols. The antibodies used were: Anti-P40 (ORIGENE, ZM-0406, working fluid), Anti-cytokeratin 5/6 (ORIGENE, ZM-0313, 1:200), and Anti-P63 (ORIGENE, ZM-0071, working fluid). Images of H&E and immunohistochemistry were acquired using the 3DHISTECH Panoramic SCAN system and analyzed using Image J software.

### 2.9. Drug Sensitivity Test on the SMAR-Chip

A drug sensitivity test on the SMAR-chip was performed following the procedures developed in our other study [22]. The viability of the LCOs was measured both before and after the addition of the anti-cancer drugs using the alamarBlue™ Cell Viability Reagent (Invitrogen, Carlsbad, CA, USA). After culturing on the SMAR-chip for 3 days, the medium was removed, and 800 nL of 10% alamarBlue reagent was added onto the Matrigel droplets. Then, the chip was incubated for 2 h in a 37 °C incubator. A slide covered on the microwell array to flatten the top of the droplets in the microwells in order to eliminate the variation in the fluorescent signal in the microwells. After that, the SMAR-chip was scanned, and the fluorescence signal was measured using an Olympus IX83 inverted fluorescence microscope. The fluorescent intensity of each microwell was measured using Image J software. Next, the alamarBlue solution was removed and the chip was submerged in culture medium for 4 h in order to completely wash away the residual alamarBlue inside the Matrigel. Next, 2.4 μL of fivefold serial diluted drugs were added in each microwell. The concentration ranges were 0.002–15.625 μM for AMG510 and 0.02048–1600 μM for doxorubicin, respectively. To eliminate the background noise introduced by the alamarBlue reagent itself, Matrigel without LCOs was used as a negative control where the alamarBlue reagent was added and the fluorescent intensity was measured. After 3 days of drug treatment, the post-treatment viability measurement was performed using the same procedure. The relative viability (the post-treatment viability divided by the pretreatment viability) of each condition was calculated and normalized by the vehicle control (0.1% DMSO).

A drug sensitivity test on the SMAR-chip was performed following the procedures developed in our other study [22]. The viability of the LCOs was measured both before and after the addition of the anti-cancer drugs using the alamarBlue™ Cell Viability Reagent (Invitrogen, Carlsbad, CA, USA). After culturing on the SMAR-chip for 3 days, the medium was removed, and 800 nL of 10% alamarBlue reagent was added onto the Matrigel droplets. Then, the chip was incubated for 2 h in a 37 °C incubator. A slide covered on the microwell array to flatten the top of the droplets in the microwells in order to eliminate the variation in the fluorescent signal in the microwells. After that, the SMAR-chip was scanned, and the fluorescence signal was measured using an Olympus IX83 inverted fluorescence microscope. The fluorescent intensity of each microwell was measured using Image J software. Next, the alamarBlue solution was removed and the chip was submerged in culture medium for 4 h in order to completely wash away the residual alamarBlue inside the Matrigel. Next, 2.4 μL of fivefold serial diluted drugs were added in each microwell. The concentration ranges were 0.002–15.625 μM for AMG510 and 0.02048–1600 μM for doxorubicin, respectively. To eliminate the background noise introduced by the alamarBlue reagent itself, Matrigel without LCOs was used as a negative control where the alamarBlue reagent was added and the fluorescent intensity was measured. After 3 days of drug treatment, the post-treatment viability measurement was performed using the same procedure. The relative viability (the post-treatment viability divided by the pretreatment viability) of each condition was calculated and normalized by the vehicle control (0.1% DMSO).

### 2.10. Statistical Analysis

Statistical tests were performed as indicated in the individual figure legends using GraphPad Prism 7.02 software. The data are presented as the means ± standard deviations, medians, or quartiles, as appropriate. Normally distributed variables were analyzed by Student’s *t*-tests. Results were considered significant with *p*-value ≤ 0.05.

## 3. Results

### 3.1. In Situ Cryopreservation Process

We developed an in situ cryopreservation method where LCOs were frozen on the SMAR-chip, ready for the subsequent drug sensitivity test. As shown in Figure 1, LCOs suspended in Matrigel solution were inoculated into the microwells, followed by cryopreservation of the whole chip, which can be stored in liquid nitrogen for a long time. To perform the drug sensitivity test, the chip was removed from liquid nitrogen and thawed by placing it in a 37 °C incubator. After a short period of culturing, drugs were delivered into the microwells and the responses of LCOs to the drugs were measured. The whole freezing and thawing process eliminates the centrifugation and resuspension steps required in conventional cell freezing methods and injury to the LCOs due to these steps.

The SMAR-chip with a 12 × 9 microwell array was fabricated, which was composed of a polycarbonate substrate with the microwells (1 mm in diameter, 200 µm in depth, and 1.25 mm in pitch) and a layer (~100 µm thick) of superhydrophobic material on the top surface of the substrate (Figure 2a,b). In order to prevent the paint from entering the microwells, a circular rim was fabricated around each microwell (Figure 2c,d). The contact angle and the SEM images of the superhydrophobic material are shown in Figure 2e,f. Droplet arrays were generated on the SMAR-chip due to the repelling effect of the superhydrophobic layer to the aqueous solution, ensuring the formation of isolated liquid conditions in the individual microwells, thus avoiding cross contamination [19]. As shown in Figure 2g, the reagents in the microwells could be changed either as a whole by the submerge–aspirate method or individually by the spot-cover method to ensure unique liquid conditions in each microwell [19]. We cultured 293T cells on the chip and evaluated cell viability using the Calcein AM/PI assay. High survival rates (94 ± 3.9%), comparable to the conventional six-well plate (95 ± 2.7%), were achieved (Figure 2h,i).

### 3.2. Vitrification Is Suitable for LCOs

In order to find the cryopreservation method suitable for organoids, we first compared slow freezing and vitrification on their effect on the phenotype and viability of lung cancer organoids. As a demonstration, a previously established lung cancer organoid line was employed. The LCOs were first frozen in conventional cryovials using both methods, then thawed and analyzed 24 h later. As shown in Figure 3a, spheroid-like morphology similar to the unfrozen organoids was observed in both groups. Tracing of the organoids at different time points (day 3, day 5, day 9 and day 13) demonstrated continuous growth in both groups without significant differences in growth rates (Figure 3a,b). Immunohistochemical staining of Ki67 detected proliferating cells in LCOs of both groups (Figure 3c). In addition, H&E staining revealed that LCOs which underwent the freeze–thaw cycle retained the 3D structure of the original organoid line (solid sphere without lumen, Figure 3c). We also investigated whether the freeze–thaw cycle affected the expression of tumor cell markers. Immunohistochemical staining indicated that LCOs in both groups retained the expression of squamous cell lung cancer markers, including p40, p63, and CK5/6 (Figure 3c). In addition, we measured drug responses of the vitrified and the slow frozen organoids to the chemotherapeutic drugs gemcitabine and cisplatin. LCOs in both groups showed resistance to the two drugs (Figure 3d), consistent with our previous results. These results indicate that cryopreservation had little effect on the phenotype of the LCOs.

Then, we analyzed the viability of the LCOs after the freeze–thaw cycle. Interestingly, the slow freezing group showed excessive cell death in the core region while the vitrified group showed relatively mild cell death (Figure 4a,b), consistent with previous reports, suggesting that the high concentration of CPA in the vitrification method protected the core of organoids from ice injury [15]. We next investigated whether there are differences in the expression of apoptosis genes and oxidative stress-related genes. The Bcl-2 family consists of a number of proteins which play important roles in the regulation of apoptosis, either functioning as promoters (such as Bid, Bax) or inhibitors (such as Bcl-XL, Bcl-2) [24]. Furthermore, the Bax/Bcl-2 ratio was regarded as an indicator of cell susceptibility to apoptosis [25]. As well as the Bcl-2 family, p53 is also known to initiate apoptosis in mammalian cells. RT-qPCR of these genes showed that the levels of apoptosis promoters and indicators (Bid, Bax/Bcl-2, p53) were significantly upregulated, while the anti-apoptosis gene Bcl-XL was downregulated in the slow freezing group (Figure 4c), consistent with increased cell death. Furthermore, the higher expression level of superoxide dismutase 1(SOD1), a stress marker which was reported to be upregulated after the freeze–thaw procedure [26], suggests that cells might suffer more from oxidative stress in the slow freezing group. In addition, the protein levels of Bcl-2 and Bcl-XL were higher in the vitrification group compared to the slow freezing group, as suggested by the peaks of the flow cytometry results (Figure 4d). These results indicate that vitrification caused less injury to the organoids compared to slow freezing.

### 3.3. On-Chip Vitrification of LCOs

Then, we demonstrated the feasibility of in situ vitrification of LCOs on the SMAR-chip. As shown in Figure 5a, organoids suspended in Matrigel were loaded into the microwells and incubated briefly. Then, the equilibration solution was delivered onto the Matrigel droplets. Following 5 min of incubation, the equilibration solution was removed, and the vitrification solution was added. Then, the chip was sealed and placed directly into liquid nitrogen for long-term storage. For thawing, The SMAR-chip was removed from the liquid nitrogen and placed in a 37 °C incubator, followed by delivery of the thawing solution and washing with culture medium. As shown in Figure 5b,c, no significant difference of proliferation capacity was observed between the off-chip and on-chip vitrification groups. Excessive cell death was not found in either groups, as indicated by the live-dead assay (Figure 5d). These results suggest that similar to vitrification in the cryovials, the on-chip vitrification can ensure high cell survival rates after the freeze–thaw procedure.

Next, we demonstrated that the responses of the LCOs to anti-cancer drugs were not changed after the on-chip vitrification–thawing cycle. After thawing, the LCOs were cultured for 3 days to recover. Then, the viability of the organoids before drug treatment was measured on the SMAR-chip using the alamarBlue reagent, followed by incubation with the drugs for 72 h. Then, the viability of LCOs post-drug treatment was measured. Organoids were treated with different concentrations of commonly used chemotherapeutic drugs gemcitabine, pemetrexed, or doxorubicin. As shown in Figure 6, both the unfrozen control and the on-chip vitrification LCOs showed resistance to gemcitabine and pemetrexed, while treatment with doxorubicin led to a dose-dependent decrease in viability in a similar manner for the two groups. In addition, because the LCOs harbored the KRAS G12C mutation, the drug AMG510, an inhibitor to KRAS G12C, was also tested. Our results indicated that the LCO was not sensitive to AMG510, echoing the diverse effect of AMG510 on cell lines harboring the mutation. For instance, H358 is sensitive to AMG510 while H2122 is resistant to it, although both cell lines have the KRAS G12C mutation. Overall, these results suggest that the in situ vitrification method enabled ready-to-use drug screening of LCOs without compromising the viability or changing the drug responses of the organoids.

## 4. Discussion

Patient-derived organoids recapitulate the genetic and structural features of parental tumor tissues, represent patient’s response to anti-cancer drugs, and are recognized as a promising model to overcome the limitations of cancer cell lines. Due to the limited proliferation capacity and heterogeneity of PDOs, new platforms enabling high-throughput organoid culture and drug sensitivity tests are essential for their future application in anti-cancer drug development. In our other study, we demonstrated that the SMAR-chip is suitable for lung cancer organoid culture and drug sensitivity tests [22]. In this study, we developed an in situ LCO vitrification method on the SMAR-chip. The whole freeze–thaw procedure can be performed with simple steps, eliminating the centrifugation and resuspension procedures, minimizing freeze injury to the LCOs. More importantly, the cryopreserved chip is ready for the subsequent drug sensitivity test, facilitating future application of PDOs in high-throughput drug screening.

We found that vitrification is better than slow freezing for the cryopreservation of PDOs. The low-concentration CPA used in the slow freezing process (usually 10% DMSO) has difficulty entering the central area of the organoid, which may cause the structure of the organoid to be destroyed due to crystal formation [15]. Previous study has shown that if organoids are cut into small pieces before cryopreservation, cell viability is increased significantly owing to the full penetration of DMSO into cells within the core [27]. The high-concentration CPA used in vitrification ensures rapid penetration into the central area of the organoids, reducing freeze injury to the cells. We observed that the number of dead cells and the expression of apoptosis indicator genes were all significantly decreased in the vitrificated LCOs compared to organoids which underwent slow freezing, consistent with previous reports [15].

Successful vitrification requires high cooling and warming rates to prevent the formation of ice crystals which can cause fatal damage to the cells. Vitrification of cells in microscale fluid volumes has been one approach to increase cooling rates [28,29]. The in situ chip cryopreservation system adopted the SMAR-chip where tiny CPA droplet arrays with a volume of 2 μL were generated, ensuring rapid and uniform temperature change in the droplet array. Another benefit of rapid cooling is that the concentration of vitrification reagent can be reduced to avoid cell toxicity [30]. In this study, we used commercial vitrification regents which performed well in our system. In the future, the components of the vitrification reagents can be optimized to further reduce cell cytotoxicity.

In our previous study, we fabricated a superhydrophobic layer of poly (propyl methacrylate) on a glass slide, and then transferred the superhydrophobic layer to a PDMS microwell array chip by a polymer transfer process, named “micrografting” [19]. In this study, we adapted a superhydrophobic paint (SHP) composed of the 1H, 1H, 2H, 2H-perfluorooctyltriethoxysilane-coated nanoparticles, which could be coated onto the plastic substrate by simple delivery and drying steps. Compared to the previous method, the new chip is easier to fabricate and more friendly to the end user. We cultured 293T cells and lung cancer organoids on the new chip and observed similar cell viability compared to cells cultured in conventional tissue culture plates. However, the biocompatibility of the chip needs further study. For example, culturing more types of cells and analysis on gene expression are needed.

In summary, we developed an in situ vitrification method on the SMAR-chip for the cryopreservation of patient-derived organoids and demonstrated that lung cancer organoids maintained viability and structural integrity after the freeze–thaw cycle. More importantly, the sensitivity of LCOs to anti-cancer drugs was consistent before and after the on-chip vitrification. Our SMAR-chip-based culture system combined with in situ cryopreservation technology can serve as a convenient tool for PDO-based drug development. In the future, an automated reagent delivery system will be developed to work with the microwell array chip. These technologies will potentially facilitate the application of PDOs in anti-cancer drug development.

## Figures and Tables

**Figure 1 micromachines-12-00624-f001:**
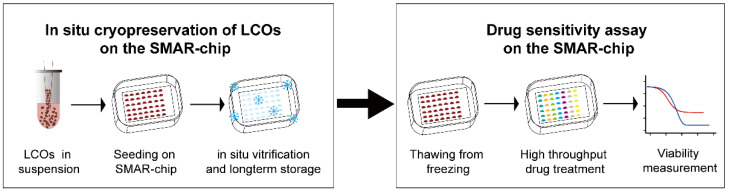
Diagram of the in situ freeze–thaw cycle and the subsequent drug sensitivity test on the SMAR-chip.

**Figure 2 micromachines-12-00624-f002:**
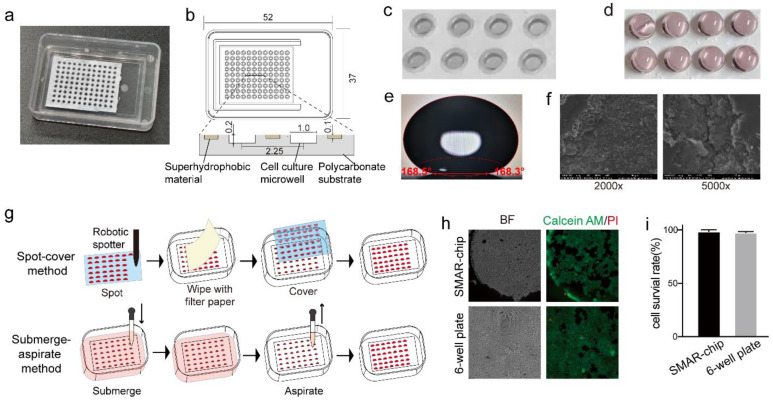
Illustration of the SMAR-chip. (**a**) The macroimage of the SMAR-chip. (**b**) The dimension of the SMAR-chip (unit: mm). (**c**) Image showing the microwell and the surrounding superhydrophobic layer. (**d**) Image of droplets in the microwells. (**e**) Contact angle of the superhydrophobic paint. (**f**) SEM images of the superhydrophobic paint. (**g**) Methods for reagent delivery on the SMAR-chip. (**h**) Images of the live/dead assay. (**i**) Quantification of cell survival rate.

**Figure 3 micromachines-12-00624-f003:**
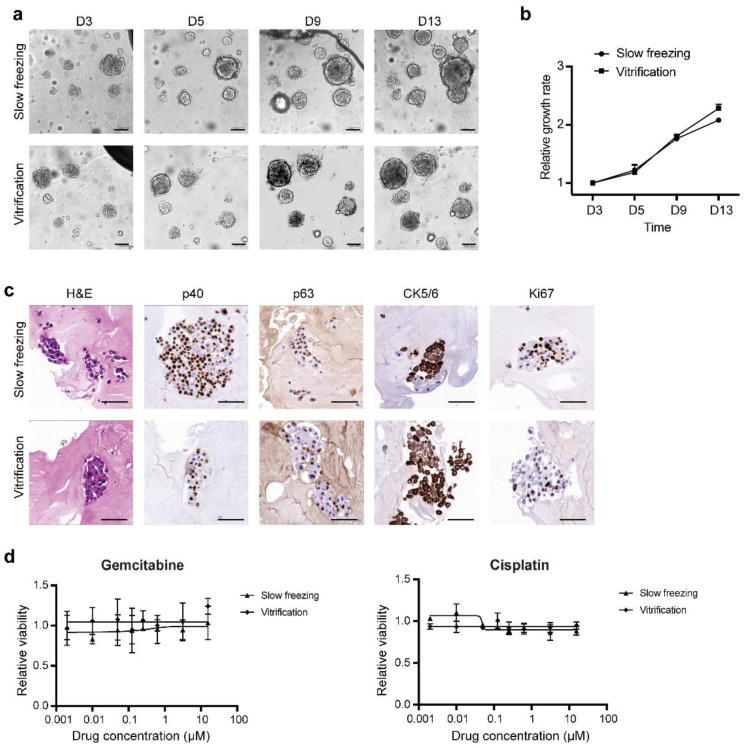
Both the slow freezing and the vitrification methods retained the morphology and cancer marker gene expression. (**a**) Tracing of LCOs thawed from slow freezing and vitrification. The LCOs were cryopreserved using conventional cryovials. (**b**) Growth rates of the LCOs in the slow freezing and the vitrification groups. The growth rates were quantified by measuring the area of the LCOs (data are represented as mean ± SD). (**c**) The histological and immunohistological staining of LCOs in the slow freezing and vitrification groups. (**d**) Drug responses of the LCOs underwent slow freezing and vitrification (data are represented as mean ± SD). Scale bars in (**b**,**c**): 100 µm.

**Figure 4 micromachines-12-00624-f004:**
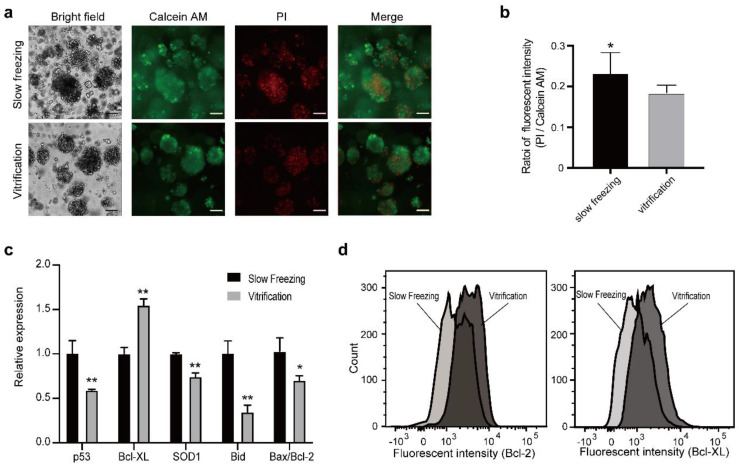
LCOs in the vitrification group maintained higher viability compared to LCOs in the slow freezing group. (**a**) Live/dead cell staining of LCOs in both groups (green: live cells, red: dead cells). Scales bars: 100 µm. (**b**) Quantification of the live/dead assay (two-tailed Student’s *t*-test, data are presented as mean ± SD, * *p* < 0.05). (**c**) RT-qPCR analysis of apoptosis-related genes and oxidative stress-related genes (two-tailed Student’s *t*-test, data are presented as mean ± SD, * *p* < 0.05, ** *p* < 0.01). (**d**) Flow cytometry measurement of the anti-apoptosis genes Bcl-2 and Bcl-XL.

**Figure 5 micromachines-12-00624-f005:**
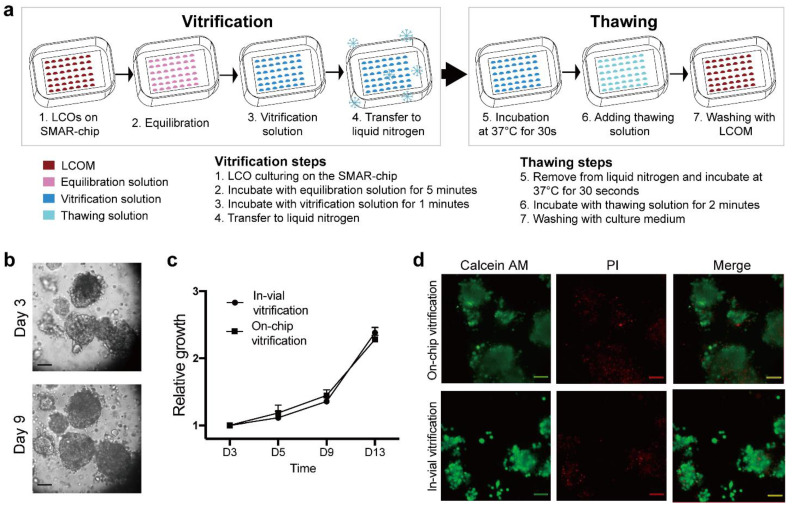
Illustration of the in situ vitrification-thaw method. (**a**) Diagram showing the procedures of the in situ vitrification–thawing cycle on the SMAR-chip. (**b**) Images of the LCOs after the in situ vitrification–thawing cycle. (**c**) Growth rates of LCOs underwent the vitrification–thawing cycle on the SMAR-chip and in the cryovial. (**d**) Images of the live–dead cell staining of LCOs underwent the on-chip and in-vial vitrification. Scales bars: 100 µm.

**Figure 6 micromachines-12-00624-f006:**
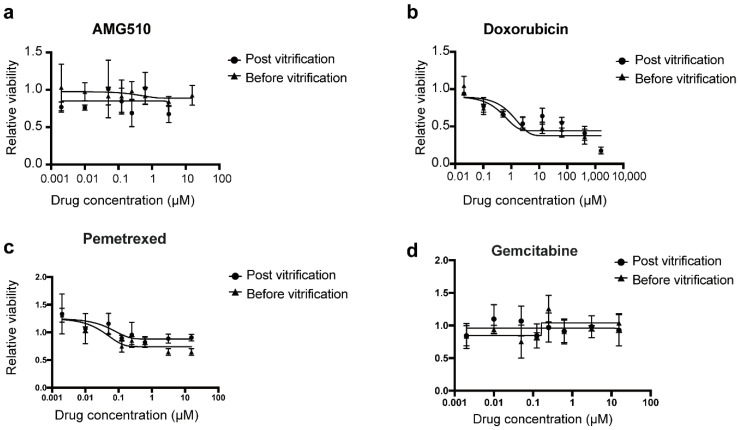
Drug sensitivity of the LCOs before and after the on-chip vitrification–thawing cycle. Drug response curves of AMG510 (**a**), doxorubicin (**b**), pemetrexed (**c**), and gemcitabine (**d**) are shown.

## Data Availability

All the data of this research are included in the manuscript and the Appendix A and are available from the corresponding author.

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
