# Peer review of "In Situ Vitrification of Lung Cancer Organoids on a Microwell Array"

_micromachines, 2021, doi:10.3390/mi12060624_

Round 1
Reviewer 1 Report
The paper is well-constructed, however, there is a critical weakness that leads to failure of publication.
Line 82 "The SMAR-chip was fabricated by casting a layer of superhydrophobic paint on the top of the polycarbonate microwell array-chip which was manufactured by the standard injection molding. The superhydrophobic paint was prepared following Lu’s protocol[22]."
1H, 1H, 2H, 2H-perfluorooctyltriethoxysilane (Sigma-Aldrich, St. Louis, 85
USA) is NOT a biocompatible chemical compound Please check below SDS from Sigma website
https://www.sigmaaldrich.com/MSDS/MSDS/DisplayMSDSPage.do?country=US&language=en&productNumber=667420&brand=ALDRICH&PageToGoToURL=https%3A%2F%2Fwww.sigmaaldrich.com%2Fcatalog%2Fproduct%2Faldrich%2F667420%3Flang%3Den
In our lab, we use this chemical a lot for surface hydrophobic coating.
Reference 22, is the initial invention of the method, It has over 1000 citations so far, however, I checked the 1st 12 pages on google scholar, I could not find even one cell biology application using this coating method.
Reference 19-21 from the author's previous works. I could not find a single word of "silane" from these three papers, which means the authors never used this fabrication method before. Therefore I doubt if this method works theoretically.
So please provide the SEM images of the coated surface with low and high magnifications.
Please provide a coating result showing good hydrophobicity in terms of contact angles, and compare it with a control method.
Reviewer 2 Report
This work from Dr. Liu and coworkers is timely and interesting. The authors envisage a novel method for organoid freezing by using vitrification. Rapid-cooling technologies and vitrification for oocytes and embryos are extensively accomplished in ART and best practices are already discussed in public guidelines and official documents. As well, other protocols have been published on vitrification for cryopreservation of 2D and 3D stem cells culture. While the in situ vitrification technology may eliminate the harvesting and centrifugation steps needed during conventional cryopreservation, apparently preserving viability and drug response of the thawed structures, we believe that additional experiments and experimental details would be needed to fully support the authors conclusions regarding the similarity of the LCOs slowly frozen vs the vitrified ones in terms of viability and drug response.
In particular:
Fig. 4. It would be desirable to have a quantitation of the cell death, rather than representative pictures as in 4a, when comparing vitrified vs slowly frozen organoids.
The differences in pro apoptotic vs antiapoptotic genes shown in fig. 4b do refer to mRNA levels. It is unclear, only from the data shown in fig. 4b, whether the difference in mRNA levels translates into different protein levels and whether this impinges significantly on the viability of the compared vitrified vs slowly frozen organoids. The proliferation data shown in fig. 3b would argue against a significant difference in cell viability. The possibility exists that those mRNA fluctuations may be the effect of exposure to high serum/DMSO and they may be transient. The authors should kindly provide western blotting and/or quantitative immunofluorescence for apoptosis effector proteins, to support the idea of decreased viability into the slowly frozen group.
Fig.6. We wish the authors used a larger panel of drugs to test the response of fresh vs thawed LCOs. Would the authors explain why the choice of AMG510 for their testing? Is there a biological reason or previously obtained data to support such a choice? Additionally, we believe it would be useful to compare (for drug responsiveness) slowly frozen (and thawed) LCOs with the vitrified ones. This would allow more robust conclusions regarding the biological similarity of the two systems analyzed and would aid understanding how the vitrification process, as compared to the conventional freezing, relates to the drug responsiveness.
The text may need further revision (e.g: lines 74-75)
Reviewer 3 Report
Though I was not able to access the supplementary Tables (S1 & S2), the manuscript is well written with all appropriate controls in place. Few grammatical/typos present.
Round 2
Reviewer 1 Report
The feedback from authors addressed part of my concerns. Believe are some issues to be fixed before it can be published
1. As long as the coating layer directly contacted with cell culture media, there can be a potential cytotoxicity issue if the coating material is not biocompatible. As the authors mentioned there are protection rims between the actual cell culture wells and the coating layer, It is better to give a brief description of how these rims are fabricated, and also provide images showing the physical separation of the well, rim, and the coating area. I can not see any surrounding rim structure in the provided images. It is more like an SHP layer coating on polycarbonate.
2. Line 227,"the repelling effect of the superhydrophobic layer to the aqueous solution, ensuring the formation of isolated liquid conditions in the individual microwells, thus avoiding the cross-contamination". Based on my understanding, the only function of the SHP is to provide a repelling area and minimize the cross contamination issue. However, as the authors mentioned the pumped-up rim structures can also do the work, why there is still a need for this coating? Besides, the title superhydrophobic microwell is misleading, it is actually microwell arrays with SH coated spacing/protecting area rather than wells. I suggest changing the title to clarify this point.
3. This SHP microwell array method has been published in few journals by the same author, please compare your previous work to clarify the new and better as well as limitations of the described fabrication material and method in this work.
4. Is LCOs adhesive? Please explain why the LCOs do not leak out from the wells when the well array immersed into liquid nitrogen?
Reviewer 2 Report
The authors satisfied all the criticisms raised and supplemented the work with satisfactory results for its greater completeness and strength.
I think the job is now acceptable.
Author Response
Thanks for the positive feedback.
Round 3
Reviewer 1 Report
The feedback from the authors has addressed my questions, I would like to recommend this work to be published on Micromachines